# A Stepwise Screening Protocol for Multiple Myeloma

**DOI:** 10.3390/jcm12041345

**Published:** 2023-02-08

**Authors:** Marta Morawska, Jadwiga Dwilewicz-Trojaczek, Tomasz Stompór, Piotr Ligocki, Marek Stopiński, Michał Sutkowski, Norbert Grząśko, Anna Kordecka, Mariusz Kordecki, Artur Jurczyszyn, Dominik Dytfeld, Tomasz Wróbel, Krzysztof Jamroziak, Agnieszka Druzd-Sitek, Adam Walter-Croneck, Krzysztof Giannopoulos

**Affiliations:** 1Experimental Hematooncology Department, Medical University of Lublin & Hematology Department, St. John’s Cancer Center, 20-093 Lublin, Poland; 2Department of Hematology, Transplantation and Internal Medicine, Medical University of Warsaw, 02-091 Warsaw, Poland; 3Department of Nephrology, Hypertension and Internal Medicine, University of Warmia and Mazury, 10-719 Olsztyn, Poland; 4Department of Internal Diseases, Clinical Department of Rheumatology, 10th Military Research Hospital and Polyclinic IPHC in Bydgoszcz, 85-681 Bydgoszcz, Poland; 5Internal Department, Geriatric Unit, John Paul II Western Hospital in Grodzisk Mazowiecki, 05-825 Grodzisk Mazowiecki, Poland; 6College of Family Physicians in Poland, 00-209 Warsaw, Poland; 7HTA Registry, 30-552 Krakow, Poland; 8Plasma Cell Dyscrasia Center, Department of Hematology, Faculty of Medicine, Jagiellonian University Medical College, 31-007 Krakow, Poland; 9Department of Hematology, Poznan University of Medical Sciences, 60-512 Poznan, Poland; 10Department of Hematology, Blood Neoplasms and Bone Marrow Transplantation, Wroclaw Medical University, 50-556 Wroclaw, Poland; 11Department of Lymphoid Malignancies, Maria Sklodowska-Curie National Research Institute of Oncology, 00-001 Warsaw, Poland; 12Department of Haemato-Oncology and Bone Marrow Transplantation, Medical University of Lublin, 20-081 Lublin, Poland

**Keywords:** monoclonal gammopathy, multiple myeloma, screening

## Abstract

Background: Monoclonal gammopathies and multiple myeloma should be screened in the primary care setting. Methods: The screening strategy consisted of an initial interview supported with the analysis of basic laboratory test results and the increasing laboratory workload in the following steps was developed based on characteristics of patients with multiple myeloma. Results: The developed 3-step screening protocol includes evaluation of myeloma-related bone disease, two renal function markers, and three hematologic markers. In the second step, the erythrocyte sedimentation rate (ESR) and the level of C-reactive protein (CRP) were cross-tabulated to identify persons qualifying for confirmation of the presence of monoclonal component. Patients with diagnosed monoclonal gammopathy should be referred to a specialized center to confirm the diagnosis. The screening protocol testing identified 900 patients with increased ESR and normal level of CRP and 94 of them (10.4%) had positive immunofixation. Conclusions: The proposed screening strategy resulted in an efficient diagnosis of monoclonal gammopathy. The stepwise approach rationalized the diagnostic workload and cost of screening. The protocol would support primary care physicians, standardizing the knowledge about the clinical manifestation of multiple myeloma and the method of evaluation of symptoms and diagnostic test results.

## 1. Introduction

Multiple myeloma screening is not routinely performed. There is no method available that is sensitive, specific, and cost-effective. However, the morbidity of multiple myeloma is significant if detection is delayed [1,2]. Delays in the diagnosis of myeloma result from non-specific symptoms of the disease. Patients frequently visit multiple physicians before diagnosis, and a recent study showed that the most common route of diagnosis was the emergency department [3]. Patients with multiple myeloma experience the longest interval from initial symptom reporting to diagnosis among all patients with common cancers [4,5]. Myeloma is an uncommon cancer; thus, any screening would lead to many unnecessary tests. General practitioners commonly see symptoms of myeloma-related end-organ damage described by the acronym CRAB: hypercalcemia, renal failure, anemia, and bone lesions. Most frequently, they cannot observe biomarkers of organ failure, which updated diagnostic criteria in 2014 to enable early diagnosis. These criteria were described by the acronym SLiM: clonal bone marrow plasma cells greater than or equal to 60%, serum-free light chain (FLC) ratio greater than or equal to 100 provided involved FLC level is 100 mg/L or higher, or more than one focal lesion on MRI [6,7]. Due to the rarity of myeloma, primary care physicians usually do not consider the diagnosis. However, they would have a large potential to suspect myeloma when aware of signs that may indicate the disease.

Here, we aimed to develop a diagnostic calculator that would assist general practitioners in identifying patients with suspected myeloma. The ideal situation would be to follow the American Academy of Family Physicians guidelines based on a wide range of initial and confirmatory tests [8]. Unfortunately, in Poland and many European countries, broad access to laboratory diagnostics is limited [9,10]. Thus, effective screening needs to consider not only disease- and patient-related factors but also the reality of the health care system. The stepwise screening, limiting the number of patients at each step, would decrease diagnostic workload and keep the most expensive diagnostic procedures for highly selected patients. We aimed to develop a screening strategy that would first use the interview method combined with the review of results of basic diagnostic tests. In the second and third steps, the diagnostic workload would expand, identifying the group of patients highly suspected to have multiple myeloma or at risk of its development. The proposed method aimed to limit a patient burden, unnecessary diagnostic tests, and cost of the screening.

## 2. Materials and Methods

The study was divided into two parts: design and practical use. In the design part, a stepwise screening method was developed to allow the referral of patients from primary healthcare to specialized care to confirm the diagnosis of multiple myeloma. In the validation part, primary care physicians used the developed screening protocol in real-world practice to identify patients with suspected multiple myeloma.

### 2.1. Design of the Screening Method

We searched the literature and identified the most frequently occurring symptoms and levels for biochemical parameters. We confronted the outcomes of this review with clinical data from eight centers in Poland. In March 2017, we collected clinical outcomes from medical records of 605 patients diagnosed with MM (information about age, sex, presence of symptoms, and results of laboratory tests at diagnosis of MM). The data from the literature review and real-world clinical data at diagnosis were used to design the screening method. The first step of screening was designed to be based on an interview with a patient and the results of the most basic diagnostic tests. After referral to the second step, the results of additional diagnostic tests were required. Based on their results, patients would be referred to perform confirmatory tests.

### 2.2. Step 1. Symptom and Basic Laboratory Tests Result

In the first step, we aimed to design the scoring system based on patient-reported outcomes and laboratory tests. Qualifying patients perform additional diagnostic procedures in the second step.

The number of symptoms in the studied population was evaluated to design the symptoms scoring system. All symptoms more frequent than present in 20% of the studied population were included in the analysis. Questionnaire items were developed from the quantitative data, the relative weighting of the frequency of symptoms was based on percentages of affirmative responses. In addition, clinical relevance or importance and previous qualitative research were considered during item weighting. To calculate the final score, the presence of each questionnaire item was multiplied by the adjusted weight and the sum of points was obtained. Additional points would be obtained in the presence of up-to-date results of basic laboratory tests such as the total blood count and creatine. Because of the high diagnostic value of normocyte anemia, the scoring included the correlation of hemoglobin concentration and mean corpuscular volume (MCV). This resulted in a higher score with more severe anemia and normal MCV. Creatinine concentration was added to the scoring system to consider those before organ damage according to the SLiM criteria [6].

Symptom-based scores alone or with additional diagnostic test results scores were applied to the database containing patient records to determine the eligibility threshold for the step two screening. The minimal score threshold was set at 80% of patients. It was estimated that CRAB features are present in 80% of patients [7]. Moreover, we analyzed frequencies of the commonly coexisting symptoms.

### 2.3. Step 2. Extended Laboratory Results-Based Scoring

The second step of screening aimed to identify patients with suspected MM and perform confirmatory tests of electrophoresis and immunofixation. This included an analysis of results of the erythrocyte sedimentation rate (ESR) and concentration of the C-reactive protein (CRP). Intervals of measures of the tests were incorporated into a contingency table to describe three cross categories of ESR and CRP levels based on the literature: necessary or possible referral for immunofixation test or lack of rationale for such the referral.

### 2.4. Step 3. Confirmatory Test

In step three, referred patients had electrophoresis and immunofixation. Patients with abnormal serum immunofixation were referred to tertiary centers to assess if diagnosed gammopathy had a clinical significance.

### 2.5. Practical Use of the Screening Protocol

The screening protocol was tested in the primary care setting. Physicians obtained electronic screening calculators to conduct structured interviews with patients and read the result of screening. Doctors used step one screening at their discretion and patients with positive results were referred to perform ESR and a concentration of CRP tests. These tests were performed at the discretion of physicians. Only step three of the screening program was monitored to assess a proportion of patients with monoclonal gammopathy among patients with positive step two screening. The cost of tests performed in steps two and three were reimbursed as part of the study.

## 3. Results

### 3.1. Design of the Screening Method

A retrospective analysis of 605 case records of patients with multiple myeloma was undertaken to assess the symptoms and laboratory test results present at diagnosis. Each record belonged to a patient who was diagnosed with multiple myeloma in one of eight participating hospitals. Characteristics of the study population are shown in Table 1.

Taking into account symptoms and test results relevant to diagnosis from a review of these case notes and the literature review, outcomes were weighted to define the level of importance. Analyzed outcomes included: the presence of skeletal pain, bone lesions, bone fractures, pathological fractures, compression fractures, weakness, proteinuria, weight loss, frequent infections (≥2/year or ≥5/year requiring antibiotic therapy), neoplastic disease in 1st-degree relatives, MGUS in 1st- or 2nd-degree relatives, and transfusions of red cell platelets concentrates. Table 2 lists the frequencies of outcomes observed in the studied population. Outcomes with frequencies over 20% were used to develop step one scoring. The set of questions was compiled to enable patient assessment in direct interviews. Pathological and compression fractures were combined into a single item since they are frequently not differentiated by patients. The presence of typical traumatic fractures and neoplastic disease in 1st-degree relatives were removed based on experts’ opinions due to limited specificity. Weights were assigned based on the statistical analysis of the dataset and then adjusted based on the literature review and expert knowledge (Table 2).

The weight for skeletal pain was decreased since it is one of the most common complaint in primary care [11]. Simultaneously, we increased the importance of the presence of bone lesions and a history of pathological and compressive fractures, which manifest in myeloma-related bone disease [7]. Raw weighted 5-item symptoms scoring resulted in the total sum of 10 points and 13 points after adjustment. Depending on the presence of results of diagnostic tests, additional points were added. Table 3 presents a schema for score calculation based on hemoglobin concentration, MVC, and creatinine level.

Anemia (Hb < 12 g/dL) of different severity was detected in 382/604 patients (63.2%). The highest score (score = 2) was attributed to the cease of the severe anemia with the concentration of Hb 6.5 g/dL. In 87.4% of patients (334/382), anemia was normocytic with MVC between 80 and 100 fL. Serum creatinine was increased over >1.3 mg/dL in 139/596 patients (23.3%). The individual scores related to the concentrations of hemoglobin and creatinine were increasing with the increase in deviation from the normal value (score = 0). The final hemoglobin score was derived by multiplying the estimate by 2 or 1, depending on the value of MCV. With access to the test results, physicians can add a maximum of 5 additional points to the symptom-based score. The maximum score in step one is 18.

To define the threshold score, which qualifies patients for the second screening step, we calculated scores for patients in the studied population. The cut-off point was set at 4; 80.3% of patients with multiple myeloma had a minimum of 4 points when only a 5-item symptoms screening panel was used and 87.2% when the laboratory test-based screening part was added.

The second step of screening consisted of broadening the diagnostic spectrum. Tests included ESR and concentration of CRP. These tests were not routinely performed at diagnosis of myeloma. Only 258/605 (42.6%) and 398/605 (65.8%) patients had documented results of ESR and concentration of CRP in the studied population, respectively, and were least common compared with other laboratory tests (Table 1). ESR and plasma viscosity have demonstrated value as rule-in tests previously [12]; level CRP itself was considered as a parameter with limited value for myeloma diagnosis and monitoring [12,13]. Increased levels of CRP are present in many diseases. However, they were not correlated with ERP before. Based on the results obtained, we decided to refer patients to the next diagnostic step based on the correlation between both parameters. In the studied population, over half of the patients (200/398) had a normal level of CRP (<3.1 mL/dL). Table 4 shows the correlation matrix used for the referral to the third step. It included immunofixation aiming to diagnose monoclonal gammopathy and referral for diagnostics in tertiary centers to confirm or exclude the diagnosis of myeloma. In the studied population, 81.8% of patients (438/535) had the M component peak at classical electrophoresis.

### 3.2. Practical Use of the Screening Protocol

The developed stepwise screening protocol was tested in the primary care setting. Between May 2017 and January 2018, 140 primary care physicians from 38 healthcare organizations located in four Polish voivodeships were using the developed diagnostic calculator in their everyday practice. The total number of screened patients is unknown since steps one and two of the screening were performed at the physician’s discretion. Based on the analysis of ESR and the level of CRP at the second step, 900 patients were referred to perform immunofixation. The positive result of immunofixation was obtained in 94 cases (10.4%). All the patients were referred to specialized centers with suspicion of multiple myeloma.

## 4. Discussion

We presented the clinical characteristics of Polish patients with multiple myeloma at its diagnosis for the first time. The frequencies of myeloma-related bone disease manifestation were similar to the previous US cohort of patients from 1985 to 1998 [14]. Riccardi A et al. showed that the progress of medicine, increasing awareness of the diseases and the availability of diagnostic methods changed the manifestation of myeloma at the time of diagnosis. With time, advanced disease or emergency department diagnoses were less common [14]. However, the profile of symptoms observed in Polish patients was comparable to that observed in British patients at the beginning of the second half of the 20th century [14]. No national studies reported the time from presentation of myeloma to its diagnosis; the advanced disease status of Polish patients may suggest a very long path.

We developed a stepwise diagnostic scoring strategy to identify people highly suspected to have multiple myeloma or at high risk of its development. In the first step, patients are interviewed to evaluate the presence of the symptoms of the disease. The questionnaire contained five items: skeletal pain, bone lesions, weakness, proteinuria, and pathological and/or compressive fractures. Most of them are markers of myeloma-related bone disease but also other rheumatological or orthopedic diseases, which require a holistic view of the patient; also unusual for myeloma are patients at a young age.

Weakness is most likely associated with anemia, which was not included in the questionnaire but assessed based on hemoglobin concentration. Proteinuria is the most basic marker of renal insufficiency, present in around 30% of patients at diagnosis. The biochemical marker of renal insufficiency included in the first step of screening was the concentration of creatinine. Total blood count and biochemical assays are Poland’s most popular diagnostic tests and comprise over 85% of all tests performed in 2015–2016 [10]. A nationwide study showed that 42% of adult Poles had a minimum one total blood count performed in 2016 [15]. This justifies the addition of these two tests to the first step of screening and broadens the screening spectrum of kidney and hematologic dimensions of the myeloma. The skeletal dimension was covered only in the interview since its diagnostics require a skeletal survey. The calcium concentration was normal in most patients, making this biomarker useless for screening.

We consider the first step of the screening as a critical one. Myeloma is a disease of the elderly, and the mean number of conditions diagnosed in a patient older than 60 years of age in Poland was 3.6 [16]. The symptoms of these diseases often seen in the primary care offices fit well with symptoms of myeloma, which is hides the condition. Thus designed here, initial screening had to point the attention of a practitioner to quite usual coincidence of symptoms and a broader interview focused on the possibility of diagnosis of myeloma. The only single symptoms needed to step into future diagnosis are the presence of bone lesions and severe normocyte anemia.

Depending on the result of the interview and initial diagnostic tests, a patient can be referred to the second step of screening, which needs to perform two diagnostic tests: ESR and level of CRP. They are both in the scope of tests of primary care physicians; however, ESR is not performed frequently due to time constraints and low informational value compared with other markers of inflammation. However, the retrospective study showed a strong correlation between high ESR and normal concentration of CRP, which was confirmed in our cohort [12]. A matrix based on the results of both tests should allow to move most patients with myeloma to the next diagnostic step and help identify the group in which suspicion of the disease cannot be easily excluded without additional analysis. The false negativity rate was 8.7% and primarily concerned patients with myeloma with ESR lower than 10 mm after 1h.

Finally, suspicion of myeloma was verified at the third step with immunofixation. Unusual electrophoresis patterns are common and may be misinterpreted. Approximately 20% of patients with multiple myeloma with light chain disease have a normal electrophoresis result. Therefore, immunofixation needs to be used to detect monoclonal protein. It is justified to use the most sensitive method at the end of the screening; even the use of extra reagents significantly increases the cost [17]. Electrophoresis can be used in most of cases. Currently, immunofixation is not in the set of tests available for Polish primary care doctors.

Prospective validation at each step of the screening protocol was not performed at this step. Physicians participating in the study used the diagnostic calculator on their discretion and in case of positive screening at the second step, they had an opportunity to refer patients to immunofixation. Every tenth person referred had a positive result, indicating the presence of monoclonal gammopathy. All these persons were contacted with reference centers informed about suspicion of multiple myeloma. However, the final diagnoses were not known, patients with monoclonal gammopathy need to be followed-up due to a high risk of progression to multiple myeloma. The risk of progression from smoldering multiple myeloma to symptomatic disease is around 50% 5 years from diagnosis [18].

Proper diagnosis at an early stage of multiple myeloma is necessary for preventing related complications and providing sufficient treatment. The screening test must have sufficient sensitivity to identify as many patients as possible and be cost-effective enough to apply to the general population. The protocol described here is the first attempt to implement population-level screening based on the multiple parameters that need to be progressively increased at each step. This allows limiting the cost of screening and the number of unnecessary procedures. The proposed three-step rationalization of diagnostic workload used in the study may allow adaptation of primary care practice and focus efforts and attention on the specific group of persons with the pattern of symptoms. This would allow earlier diagnosis of multiple myeloma, monoclonal gammopathies, and other diseases that cause paraproteinemia. This is especially important in countries where patients are diagnosed late, most usually at stage III of the disease, which largely affects prognosis.

## Figures and Tables

**Table 1 jcm-12-01345-t001:** Characteristics of screening design phase participants (n = 605).

Characteristic	N	Value
Age, median (range), years	605	66 (29–93)
Males, n (%)	605	273 (45.1%)
White blood cells, median (range), ×10^9^/L	604	5.9 (1.6–39.1)
Neutrophils, median (range), mg/dL, ×10^9^/L	581	3.4 (0.0–61.4)
Hemoglobin, median (range), mg/dL	604	11.1 (1.0–37.8)
MCV, median (range), fL	600	91.2 (74.0–366.0)
Platelet, median (range), ×10^9^/L	604	210.5 (17–1262)
GFR, median (range), mL/min/m^2^	579	60 (0–221)
Creatine, median (range), mg/dL	596	0.9 (0.3–15.7)
Calcium, median (range), mg/dL	587	9.3 (0.2–221)
CRP, median (range), mg/L	398	3.0 (0–370)
ESR, median (range), mm after 1 h	258	66.0 (1–150)

CRP: C-reactive protein; ESR: erythrocyte sedimentation rate; GFR: glomerular filtration rate; MCV: mean corpuscular volume; SD: standard deviation.

**Table 2 jcm-12-01345-t002:** Frequencies of outcomes observed in the studied population (N = 605). Based on the analysis of outcomes each item was weighted and in the second step weights were adjusted based on clinical knowledge and the literature review. They were used at step 1 of the screening based on interview with the patient.

Item, n = Number of Non-Missing Records	Patients with Item Present at Diagnosis, [%] *	Statistical Analysis-Based Weighted Score	Adjusted Score
Skeletal pain, n = 596	70.64	3	2
Bone lesions, n = 588	61.73	2	4
Weakness, n = 590	56.95	2	2
Proteinuria, n = 509	45.38	2	3
Bone fracture(s), n = 592	27.36	excluded	
Pathological fracture(s), n = 593	26.81	1	2
Compression fracture, n = 596	24.16		
Neoplastic disease in relatives, n = 379	20.84	excluded	
Weight loss, n = 533	16.89	items not included into the scoring system	
Frequent infections (≥2/year), n = 535	12.71		
Transfusions of red cell concentrate, n = 595	12.61		
Frequent infections (≥5/year), n = 535	1.68		
MGUS in relatives, n = 438	1.37		
Platelet transfusions, n = 589	1.19		

* Total is more than 100% because many patients had more than 1 abnormality. MGUS: monoclonal gammopathy of undetermined significance.

**Table 3 jcm-12-01345-t003:** Additional score dependent on the presence of laboratory test results: concentration of hemoglobin, mean corpuscular volume, and hemoglobin.

Test	Result	Score	MCV (fL)	
			80–100	<80 or >100
Hemoglobin (g/dL)	<6.5	2	multiple score by 2	multiple score by 1
	6.5–10.0	1		
	10.1–12.0	0.5		
	>12	0		
Creatine (mg/dL)	>2.0	1	not applicable	
	1.3–2.0	0.5		
	<1.3	0		

**Table 4 jcm-12-01345-t004:** Erythrocyte sedimentation rate and C-reactive protein crosstabulation matrix used for referral patients to the step 3 of the screening. Number and percentage of patients in studied population with both outcomes available were shown (N = 159).

	CRP < 3.1 mg/L	CRP 3.1–10 mg/L	CRP 10–40 mg/L	CRP > 40 mg/L
ESR < 10 mm	10 (6.3%)	1 (0.5%)	0	0
ESR 10–40 mm	31 (19.5%)	9 (5.7%)	3 (1.9%)	3 (1.9%)
ESR 40–100 mm	35 (22.0%)	21 (13.2%)	10 (6.3%)	4 (2.5%)
ESR > 100 mm	12 (7.5%)	14 (8.8%)	4 (2.5%)	2 (1.3%)

Black fields indicate that lack of rational for referral, gray fields indicate conditional referral possible after additional analysis of medical history and laboratory results, and white fields indicate need of referral.

## Data Availability

The study group will consider data sharing requests on a case-by-case basis. Requests by academic study groups for de-identified patient data with the intent to achieve aims of the original proposal can be forwarded to the corresponding author.

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
