# Peer review of "A Stepwise Screening Protocol for Multiple Myeloma"

_jcm, 2023, doi:10.3390/jcm12041345_

Round 1

Reviewer 1 Report

This is an interesting analysis of a screening protocol for multiple myeloma. However, I see the following minor problems:

1) In table 1 the mean and SD of several variables are problematic without further informations on normal distribution and should be substituted by the median and range. Regarding CRP the normal value should be mentioned .

2) In the discussion on page 8, it should be mentioned that not only myeloma but also other diseases like malignant lymphomas cause paraproteins.

2) The ESR is often called wrongly ERS.

3) The first three lines of page 4 are unnecessarily and should be deleted.

4) There are some other grammatical mistakes (for example: page 3, line 133 and page 6, line 222))

Author Response

Dear Reviewer,

Thank you for your comments. We addressed them all in the following way:

Ad 1) Thank you for the comment. The data in table 1 was changed from mean +/- SD to median with range.

Ad 2) We agree with the comment and updated it in the discussion of the revised version of the manuscript.

Ad 3) We corrected the spelling. Thank you for indicating it.

Ad 4) Thank you for pointing it out. Indeed, it was part of the journal template that should have been deleted initially.

Ad 5) Thank you, we fixed it.  

All changes are tracked.

Kindest Regards,

Reviewer 2 Report

The Article Titled "A stepwise screening protocol for multiple myeloma", authored by  Marta Morawska and colleagues provides a step=by-step guide on screening for multiple myeloma. The Idea is interesting and well-presented. I have one major point to address which follows in my Comments and Suggestions for Authors.   Major concern   It is necessary to calculate the accuracy of the proposed methodology and the evaluation of cut-off value using ROC analysis or a similar statistical test. In that way, the authors would be readily calulate the sensitivity and specificity of their assay and would strengthen their arguments.   Minor concern   Table 3 would be more comprehensive to be changed into figure with standard deviation included.

Author Response

Dear Reviewer

Thank you for your comment. Unfortunately, we could not fully address your primary concern due to the project design and a lack of data. We agree that in terms of the scientific accuracy of the proposed methods, the evaluation of cut-off value using ROC analysis will be valuable, but this study has a different design. This study aimed to provide a tool that could be used in clinical practice; therefore, we have provided some arbitrary-set cut-off values. Additionally, tests performed in steps 2 and 3 of the study were covered within the financing of this project, thereby limiting the number of patients included in each study step. Having no funding and no Ethics Committee approval beyond a certain number of patients, we could not perform a proper calculation of the sensitivity and specificity using ROC  The limitation of the study, which we indicated in the discussion, was that we do not know how many patients had confirmed diagnoses after referral. We agree that analyses proposed by the Reviewer would strengthen our arguments, but such calculations cannot be performed retrospectively at this stage.

Kindest Regards,

Round 2

Reviewer 2 Report

Given the study limitations that presented by the authors, I endorse the publication in the current form